## Perspective

behaviour

science, society, policy, trust, foresights, sustainability

**Author for correspondence:**
Stephen A. Matlin
e-mail: s.matlin@imperial.ac.uk

# Realigning science, society and policy in uncertain times

Goverdhan Mehta[1,2], Henning Hopf[2,3], Alain Krief[2] and Stephen A. Matlin[2,4]

[1]School of Chemistry, University of Hyderabad, Hyderabad 500046, India
[2]International Organization for Chemical Sciences in Development, 61 rue de Bruxelles, 5000 Namur, Belgium
[3]Institute of Organic Chemistry, Technische Universität Braunschweig, Braunschweig 38106, Germany
[4]Institute of Global Health Innovation, Imperial College London, London SW7 2AZ, UK

GM, 0000-0001-6841-4267; HH, 0000-0001-7040-6506;
AK, 0000-0002-9223-1644; SAM, 0000-0002-8001-1425

Against a backdrop of rapidly changing social, economic and geopolitical settings and ideologies, the world is facing a wide range of challenges, including in biodiversity, climate, energy, the environment, food, health and water. These can only be addressed by fully harnessing key capacities that science offers. However, there is a crisis of trust in science which affects some sections of society and some policy-makers, impairing the capacity of science to deliver its essential roles. This damaged relationship between science, society and policy has immense health, economic and social consequences and implications for sustainability of the entire planet. Scientists must strive collectively to re-establish trust by society and politicians where it is damaged, and reinforce conviction of science's central importance in underpinning policy. Science's roles must in turn be acknowledged by policies that sustain innovation and freedom to work without political interference or constraints. A well-functioning and trusting relationship between science, society and policy-makers offers a potent means to thwart and mitigate emergent global challenges.

## 1. Introduction

> Uncertainty is intrinsic to the human condition [1].
>
> Quoted from interview with French philosopher Edgar Morin.

As a source of knowledge, useful applications, solutions to problems and evidence-based advice, science and technology (S&T), including the burgeoning information and communications technologies (ICTs), are pivotal to people's quality of life and prosperity in the twenty-first century [2–4]. Critically, these capacities that science offers need to be accepted and acted upon,

which requires an ethos and approach to evaluating information that enables individuals, communities and those enacting and implementing policies at local, national and international levels to take evidence-informed decisions [5].

It is, therefore, of particular concern that the current times are characterized [6–8] as the era of 'post-truth', 'fake news' and 'fake science'. These are signals of a triple emergency: a crisis of trust within society in the reliability of the information individuals receive through multiple channels; a crisis in the credibility of politicians and a crisis within science in how it sustains its own credibility and productivity while dealing with accusations of bias, conflicted interests, hype, lack of reproducibility and falsification [9]. Moreover, increasing assaults on the independence of science include measures by some governments that are deliberately designed to impair science's capacity to maintain intellectual freedom or to present facts that contradict political positions [10]. All these developments are undermining the ability of the vital tree of knowledge that is science to continue growing and bearing fruit.

# 2. Government and society roles in reaping science's fruits

Notwithstanding all the proven and potential benefits of S&T, its advances also generate some hazards [11]. These include unethical and/or harmful practices in research, manufacturing and application, as well as unforeseen negative impacts of technologies. Consequently, it is incumbent on countries to consider the roles that governments must play with regard to S&T. One is to determine the appropriate levels of investment in science and its applications. This needs to be accompanied by policies for fostering innovation and making optimal and sustainable use of the products of S&T. Policy-makers can be stimulators and enablers of scientific research and industrial enterprise and facilitators of engagement across the critical interface between the two. This has been widely recognized in, for example, the science and innovation policies of OECD countries [12].

In addition, the investment in science needs to include cultivating the educational and intellectual infrastructure that ensures an adequate supply of well-trained scientists. It also requires the development of a sound science literacy in the general population—fostering a broad-based understanding of the potentials and limitations of science, respect for its methods and scepticism about unsupported or untested claims. Gaps in the general level of science literacy—including lack of appreciation of the time and effort needed to undertake and deliver scientific results and outcomes that are evaluated and validated by competent peers—have provided fertile ground for the growth and proliferation of untrustworthy claims and false information, especially through social media. Science literacy should be regarded as an essential life skill that will protect individuals from the harm that can be perpetrated by the false science claims and conspiracy theories propagated by, for example, 'anti-vaxxers' [9], as well as enabling citizens to assess the claims of, for example, climate change deniers [13] and to reach soundly based conclusions on critical issues on which they expect politicians to take evidence-informed policy decisions.

A further key role for the state is to determine the appropriate protections required, both for individuals and for society as a whole, against risks that may flow from unbridled profiteering and marketing, ideological motives or negligent behaviour. These protections include regulating processes and products, ensuring they are safe and not mis-applied in unethical or harmful ways. Such protections often require cross-border cooperation based on a shared, global level of understanding of and respect for scientific methods and evidence.

Many countries have long recognized the importance of these roles and responsibilities, so that using science for policy and formulating policies for science and its funding and for technology transfer, development, application, exploitation and regulation have evolved hand in hand, with society engaged as an essential third party in the relationship.

Sustaining balance in this three-way interaction is not easy [14,15]. Jealous of their autonomy, scientists will argue for more resources for 'blue-skies' research that is not fettered by constraints or targets. But politicians with an eye on the public purse tend to demonstrate control, commitment to the public good and foresight by picking areas of science in which to focus resources. And they have sometimes ignored advice they received (whether solicited or not) from scientists when the scientific evidence was at odds with their favoured policies [16]. Societal attitudes to S&T have also fluctuated in tandem with this evolving picture, with perceptions often diverging among different sections of society, for example, with regard to agriculture, environment, health, ICTs, industry and weapons of war. The accelerating speed and scale of S&T advances and the emergence of disruptive new

technologies have increasingly outpaced the general population's level of understanding, both with respect to the S&T involved and its implications. This has often led to a disconnection between science and society [17], with the technologies being adopted or rejected, welcomed or feared, more on emotional rather than evidential grounds.

To this uneasy mix has been added growing concerns about the effects on the sustainability of the very technologies that have provided the wealth, health and material benefits [18–22]. As Smil has observed [23], 'Human history is a story of innovation and increased efficiency, but also of relentless depletion of Earth's resources'. Science has provided clear, incontrovertible evidence of multiple, unfolding and mutually reinforcing sustainability crises which need to be countered with urgent, global efforts [24]. S&T have been among the causes of these crises and must be part of the solutions to them.

The case of stratospheric ozone depletion caused by chloroflurocarbons (CFCs) illustrates the extent to which many complex social, political, economic and cultural factors are involved in the way that scientific results are handled. The remarkably rapid global response, leading to the 1987 Montreal Protocol on Substances that Deplete the Ozone Layer, involved an intense interplay of many national and international factors including powerful commercial and political interests [25]. Benedick, who led the US Delegation in many of the ozone negotiations, wrote afterwards that 'To understand what was happening to the ozone layer, researchers… had to bridge traditional scientific disciplines and examine the Earth as an interrelated system of physical, chemical, and biological processes occurring on land, in oceans, and in the atmosphere—processes that were themselves influenced by economic, political, and social forces' [26]. Clear science was a necessary, but not sufficient, condition for success [27] in developing the ozone treaty.

With changes in social, political and economic conditions since the 1980s and a more gradually evolving threat to people's well-being, there has been a much slower pace of progress of the negotiations on climate change, despite clear scientific evidence for anthropogenic global warming. And, as discussed below, political and economic factors have played critical roles in many countries, including the USA, in influencing the speed, nature and extent of responses to the Covid-19 pandemic.

# 3. Changing times for uneasy but essential relationships

The Wellcome report [28] of the world's largest study into how people around the world think and feel about science and major health challenges demonstrated that levels of trust in science, as measured by survey questions, vary substantially around the world. Many factors were seen to be associated with trust levels, including learning science at school or college, confidence in key national institutions, gender, age, income, rural/urban location, level of country's income inequality, and access to mobile phones and the Internet. In addition, there could be many nuances in the attitudes of individuals. For example, people's views of science could be influenced by whether they saw it as benefitting them personally and by whether particular science contradicted their religious beliefs. However, the finding that the variables studied explained only a minor fraction of the differences found indicated that much further work is needed to better understand the factors affecting people's trust in scientists.

Against this complex and as-yet poorly understood background, evidence suggests that there has been a decline in trust in science in particular places, at particular times or on particular issues. For example, the proportion of people who asserted that vaccines are not safe was highest in the world (one-third) in France, although the general survey had found that only 9% of respondents from that country has 'low trust' in science and only 1% has 'no opinion'. Vaccination rates for measles, mumps and rubella, as well as for other diseases, have declined steeply in a number of high-income countries since false claims were circulated—and subsequently disproved—that the vaccine was associated with autism. This has led to epidemics of measles [9,29]. Increasing disconnection has been seen between science and public opinion on issues including climate change, genetic modification and evolution [30–32]. Some surveys have found a clear ambivalence of people towards science and the applications of research, which overall show distrust towards scientists' motives rather than about science itself [33].

The decline in trust and respect and sometimes undermining of the effective functioning of science have often been concurrent and interrelated with a shifting landscape of socio-economic, geopolitical and multilateral relationships, population growth and demographic changes, emerging economies and ideological and religious divides.

For example, the confluence of deglobalization, hyper-nationalism, anti-immigration and trade protectionist trends in a period of economic and environmental stresses has resulted, in a number of countries, in movements that put short-term economic interests ahead of longer-range global concerns.

Populist policies sometimes have components that are damaging to both the ethos and practice of science. Examples include the denial of facts that are contrary to political positions, coupled with cuts in funding for science as a whole, or of particular science areas that would contradict policy positions, as well as regulations contrived to prevent well-established evidence from being considered [9]. Limitation of the movement of scientists between countries leads to the narrowing of diversity and impairment of international collaborations—despite the fact that science productivity and successes, including of the USA and European countries, have depended heavily on migrating scientists and cross-border programmes up to the present time [34,35]. The rise of these inward-looking tendencies has undermined international collaborations and weakened multilateral institutions that are essential for addressing global problems through the harmonious operation of the science, society and policy interface.

Government financing of science [36] has become an area of increasing tension among scientists and of concern to society in recent years [37]. Despite the evidence that research is not a luxury but a driver of innovation and growth [38,39], governments sometimes make very large cuts that erode the science base of the country. Within the envelope of the overall science budget, allocations of funding for particular areas have always been a political choice, reflecting government priorities in some areas and restricting the autonomy of scientists in other, disfavoured areas [40,41]. The culture of science denial [42–45] displayed by some politicians and sections of society affects the approach to looming planetary disasters and also spawns a social attitude that it is acceptable to scorn rigorously proven evidence and to treat fake science and faked evidence as if it is of at least equal value for decision-making [10].

# 4. Responding to the challenges

Responses by the scientific community to the growing challenges have been mixed. Individuals, groups and institutions have published, spoken out and even taken to the streets to protest about fake science, the suppression of real science and the denigration of honest scientists [8,46–53] However, at times, responses have been subdued. Reasons that have been advanced have included the assumption that if the job of scientists is to provide information, then adopting a position will weaken their authority [54], fear of denigration, punishment or retaliation by other scientists or politicians with personal motives [55–57] and awareness that global threats involve complex processes and uncertain outcomes that can be very challenging to communicate to politicians and society [58,59], and that there are very large knowledge gaps to be overcome before messages will be taken on board [60]. With the advent of the Internet and social media, scientists have sometimes been wary of unpleasant or hateful responses if they react publicly to political stances or priorities or to the circulation of fake science.

When scientists are denigrated and essential science institutions hobbled and downsized by some governments, too many scientists quietly keep their heads down, hoping for better times when policies change. However, by behaving as silent spectators, they are enabling long-term damage to the domain of science and acquiescing to the long-term erosion of trust of society in science. This may contribute, for example, to ignoring the pressing planetary crises that will impact on the entire population of the planet in the next few decades. The remark by UN General Assembly President María Fernanda Espinosa Garcés that 'We are the last generation that can prevent irreparable damage to our planet' [61] obliges everyone, including scientists and society working together, to make their voices heard [62]. The science community must also make greater efforts to develop science literacy in the general population, fostering both an ethos of science and a respect for its methods and results that needs to be recognized as a broad and essential life skill and not as a narrow specialization. Science has also been slow in connecting with other knowledge streams, particularly social sciences and humanities [63], essential for broadening its base of support and also for enhancing cultural competencies and advancing diversity [64].

A hopeful sign can be seen in the combination and synergy of voices now being raised by sections of both society and the scientific community, regarding the emerging planetary emergencies. Some national and international leaders have taken up the challenge, while others have either only paid lip service or denied the problem or urgency of action. Not surprisingly, a frustrated young generation, alarmed at the consequences for their own futures, are becoming increasingly vocal and impatient about delays [65,66]. Thunberg has insisted [67] to deniers of the science related to global warming that 'You can't simply make up your own facts just because you don't like what you hear'.

The challenges that the world faces will not go away of their own accord. Scientists must seek alliances to strengthen their resolve and their impact—alliances both within their own communities,

**Box 1.** Science society and policy synergy—a dependable vaccine and antidote for global challenges.

Multiple global challenges are unfolding that threaten immense health, economic and environmental consequences for the whole planet. The COVID-19 pandemic is the latest—but undoubtedly not the last—example of the challenge coming from the emergence and re-emergence of infectious diseases bringing epidemics and presently untreatable infections to all parts of the world. The world proved ill-prepared for this pandemic, despite clear warnings [68]. There are many other clear threats like global water shortage, air pollution, antibiotic resistance, mental health and demographic crises for which the world is also not yet sufficiently prepared. Another example is the accelerating process of climate change that is resulting from anthropogenic sources and causing an increased number, frequency and severity of adverse weather events; and the overstepping of planetary boundaries [16] defining safe limits within which human impacts need to be contained, bringing us ever-closer to tipping points which, if not avoided, will result in major changes to the Earth's environment from which it will take millennia to recover [24].

In all these cases, science is leading in the identification of the problems, in pointing to pathways to solutions and in contributing, in combination with other knowledge streams, to credible foresight analysis. However, developing and fully implementing the solutions cannot be done by science alone. Society and politicians need to accept the facts, understand the threats, sift and weigh the projections, options and predicted outcomes and make informed choices that balance health, economic and environmental costs according to ethical principles and humane priorities.

It is only through working together that science, society and policy can serve as a dependable vaccine to prevent and prepare for the adverse impacts of emerging threats and as a potent antidote that can treat and minimize ensuing harm to health, economies and the environment. The essential glue that bonds this combination together is trust—in science, honesty and ethics.

under the umbrella of the academies and learned societies, and with societal individuals and organizations who are needed as partners. Working together for the common cause of rebuilding mutual trust and enabling the truth to prevail (box 1) will pay long-term dividends for science and society—and for the planet [69].

This compact between science and society must also recognize that uncertainty is intrinsic to the human condition and that it is necessary to expect the unexpected [1]. Science, for all its own uncertainties and continuous need to verify and sometimes revise or update its conclusions, remains the best available guide on the pathway to navigate uncertainty.

Data accessibility. This article has no additional data.

Authors' contributions. All authors contributed equally to the content of the paper and wording of the manuscript and to its review and final approval and all agree to be accountable for all aspects of its content. The manuscript was drafted and edited by S.A.M.

Competing interests. We have no competing interests.

Funding. This article was drafted at a workshop of the International Organization for Chemical Sciences in Development (IOCD) co-hosted by the University of Hyderabad and the Dr Reddy Institute of Life Sciences. It was supported by grants from the Royal Society of Chemistry, German Chemical Society and Syngenta Research and Technology Centre, Goa. IOCD is hosted at the University of Namur, Belgium.

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
