## [Reviewer comments · Royal Society Open Science]

Review History

RSOS-200554.R0 (Original submission)

Review form: Reviewer 1

Is the manuscript scientifically sound in its present form?

Yes

Are the interpretations and conclusions justified by the results?

Yes

Is the language acceptable?

Yes

Do you have any ethical concerns with this paper?

No

Have you any concerns about statistical analyses in this paper?

No

Recommendation?

Accept as is

Comments to the Author(s)

What Mehta, Hopf, Krief and Matlin write makes very good sense. Their exhortations to both science and the general public to join in giving value again to the hard-won facts of modern science are very appropriate. So publication is definitely to be encouraged.

Where this essay fails is in two ways: First, it does not deal in specifics (except for the Covid-19 mention). It's all generalities. Second it fails to pose and refer the reader to crucial questions. For instance,

1. How can it be that in a society that values technological advance (the United States) that a majority of the population does not believe in evolution?
2. Why in a country that supports science to an extraordinary extent – witness the work that led to so many US Nobel Prizes – that a large number of citizens do not believe in the results of this expertise, say on climate change?
3. Why was a previous quite technical argument on an invisible danger in ozone depletion in the atmosphere believed, leading to international legislation, but anthropogenic climate change is not believed by many?
4. Why a country as progressive in its science as Italy and Spain, countries that believed in scientific expertise, why did they not move more quickly on controlling Covid-19?

There are studies and discussions in the literature on 3 of the 4 difficult questions I raise, but this paper does not reference them. It seems to me to take the easy way out, to list the troubles, bemoan the situation, but not deal with specifics. It preaches to the choir.

Review form: Reviewer 2 (Daniël Lakens)

Is the manuscript scientifically sound in its present form?

Yes

Are the interpretations and conclusions justified by the results?

Yes

Is the language acceptable?

Yes

Do you have any ethical concerns with this paper?

No

Have you any concerns about statistical analyses in this paper?

No

Recommendation?

Accept as is

Comments to the Author(s)

I greatly enjoyed reading this perspectives piece. I think it is very strong, and I only have very minor suggestions.

Section 4 starts with "The decline in trust and respect and sometimes undermining of the effective functioning of science". However, public trust in science in general has remained stable over the last decades (e.g., <https://www.pewresearch.org/fact-tank/2019/03/22/public-confidence-in-scientists-has-remained-stable-for-decades/>). If the authors want to argue for a decline in trust in science, they will either need to support their argument with data and references, or they might want to be more specific in what they mean (e.g., a greater concern about trust in science, or more visibility of distrust in science – but general trust seems stable). The link goes to 1 survey, but I had a post-doc once collect these surveys across countries – the pattern

is basically the same everywhere. Now the questions in these surveys might be too general, and not measure the same type of trust as the authors mean. But I think some clarification is required.

Personally, I found the last sentence of section 4 too strong: "The culture of science denial displayed by some politicians and sections of society affects the approach to looming planetary disasters and also spawns a social attitude that it is acceptable to scorn rigorously proven evidence and to treat fake science and faked evidence as if it is of at least equal value for decision-making." I mean, yes, these are troublesome developments, but not unique our history. We are relatively far removed from treating fake science as a good source of decision making. The manuscript does not mention there is a huge pushback to fake news as well, so in some ways it also increases the awareness of the importance of reliable information among others, who work actively to make sure internet pages are reliable (e.g., think of hundreds of thousands of moderators cleaning up internet for a such as reddit).

In section 5, I would like to see a reference after "Responses by the scientific community to the growing challenges have frequently been muted." I wonder which examples the authors are thinking of.

Related to 2 points above, when the authors write "Scientists have often been slow to react publicly to the circulation of fake science" I feel they are not giving enough credit to the many scientists actively fighting fake news. If I see how someone like Carl T. Bergstrom is fighting fake news on twitter, I think such people deserve some recognition.

I am in doubt of the value of the figure in Box 1. I think this article is really strong. I greatly enjoyed reading it. This figure is by far the weakest part in the manuscript. It's not needed, overly simplistic, and the corona virus might be a good hook, but this perspective piece will stay relevant in future crises. I would probably take it out myself. If not, the picture might need a credit (I assume the corona virus picture was taken from somewhere).

It was a pleasure to read this perspective. I look forward to seeing it in print, and circulating it among my colleagues.

Signed,
Daniel Lakens

Review form: Reviewer 3

Is the manuscript scientifically sound in its present form?

Yes

Are the interpretations and conclusions justified by the results?

Yes

Is the language acceptable?

Yes

Do you have any ethical concerns with this paper?

No

Have you any concerns about statistical analyses in this paper?

No

Recommendation?

Accept with minor revision (please list in comments)

Comments to the Author(s)

This article highlights the importance of "trust in science", the need for a scientifically literate population and for a strong and fruitful relationship between science and policy. It emphasises the dangers of ignoring science based evidence as in climate change denial and the "anti-vaxxer" movement (which might be discussed in the article). Although all these points have been made before, they are well worth highlighting and the article should be published.

I have two concerns which I would like the authors to consider. First, I think there is an unduly pessimistic tone to the article. It refers to re-establishing trust" and whereas I accept that there are issues here, I think there is still a great deal of confidence and respect for science and scientists in many countries and is some (eg the UK), we have seen an increase in the recognition by governments of the importance of science to the economy and society generally. I think the impact of the article will be greater if these positive aspects are recognised more clearly.

Secondly, the style of the article deserves some attention. It could in parts be clearer and more readable.

Decision letter (RSOS-200554.R0)

Dear Dr Matlin:

On behalf of the Editors, I am pleased to inform you that your Manuscript RSOS-200554 entitled "Realigning science, society and policy in uncertain times" has been accepted for publication in Royal Society Open Science subject to minor revision in accordance with the referee suggestions. Please find the referees' comments at the end of this email.

The reviewers and Subject Editor have recommended publication, but also suggest some minor revisions to your manuscript. Therefore, I invite you to respond to the comments and revise your manuscript.

- Ethics statement

- Data accessibility

If you wish to submit your supporting data or code to Dryad (<http://datadryad.org/>), or modify your current submission to dryad, please use the following link:
<http://datadryad.org/submit?journalID=RSOS&manu=RSOS-200554>

- **Competing interests**

- **Authors' contributions**

- **Acknowledgements**

- **Funding statement**

Because the schedule for publication is very tight, it is a condition of publication that you submit the revised version of your manuscript before 01-May-2020. Please note that the revision deadline will expire at 00.00am on this date. If you do not think you will be able to meet this date please let me know immediately.

If your manuscript is newly submitted and subsequently accepted for publication, you will be asked to pay the article processing charge, unless you request a waiver and this is approved by Royal Society Publishing. You can find out more about the charges at <https://royalsocietypublishing.org/rsos/charges>. Should you have any queries, please contact openscience@royalsociety.org.

on behalf of Professor Matjaz Perc (Associate Editor)
openscience@royalsociety.org

Comments to Author:

Reviewer: 1

Comments to the Author(s)

What Mehta, Hopf, Krief and Matlin write makes very good sense. Their exhortations to both science and the general public to join in giving value again to the hard-won facts of modern science are very appropriate. So publication is definitely to be encouraged.

Where this essay fails is in two ways: First, it does not deal in specifics (except for the Covid-19 mention). It's all generalities. Second it fails to pose and refer the reader to crucial questions. For instance,

1. How can it be that in a society that values technological advance (the United States) that a majority of the population does not believe in evolution?
2. Why in a country that supports science to an extraordinary extent – witness the work that led to so many US Nobel Prizes – that a large number of citizens do not believe in the results of this expertise, say on climate change?
3. Why was a previous quite technical argument on an invisible danger in ozone depletion in the atmosphere believed, leading to international legislation, but anthropogenic climate change is not believed by many?
4. Why a country as progressive in its science as Italy and Spain, countries that believed in scientific expertise, why did they not move more quickly on controlling Covid-19?

There are studies and discussions in the literature on 3 of the 4 difficult questions I raise, but this paper does not reference them. It seems to me to take the easy way out, to list the troubles, bemoan the situation, but not deal with specifics. It preaches to the choir.

Reviewer: 2

Comments to the Author(s)

I greatly enjoyed reading this perspectives piece. I think it is very strong, and I only have very minor suggestions.

Section 4 starts with "The decline in trust and respect and sometimes undermining of the effective functioning of science". However, public trust in science in general has remained stable over the last decades (e.g., <https://www.pewresearch.org/fact-tank/2019/03/22/public-confidence-in-scientists-has-remained-stable-for-decades/>). If the authors want to argue for a decline in trust in science, they will either need to support their argument with data and references, or they might want to be more specific in what they mean (e.g., a greater concern about trust in science, or more visibility of distrust in science – but general trust seems stable). The link goes to 1 survey, but I had a post-doc once collect these surveys across countries – the pattern is basically the same everywhere. Now the questions in these surveys might be too general, and not measure the same type of trust as the authors mean. But I think some clarification is required.

Personally, I found the last sentence of section 4 too strong: "The culture of science denial displayed by some politicians and sections of society affects the approach to looming planetary disasters and also spawns a social attitude that it is acceptable to scorn rigorously proven evidence and to treat fake science and faked evidence as if it is of at least equal value for decision-making." I mean, yes, these are troublesome developments, but not unique our history. We are relatively far removed from treating fake science as a good source of decision making. The manuscript does not mention there is a huge pushback to fake news as well, so in some ways it also increases the awareness of the importance of reliable information among others, who work actively to make sure internet pages are reliable (e.g., think of hundreds of thousands of moderators cleaning up internet for a such as reddit).

In section 5, I would like to see a reference after "Responses by the scientific community to the growing challenges have frequently been muted." I wonder which examples the authors are thinking of.

Related to 2 points above, when the authors write "Scientists have often been slow to react publicly to the circulation of fake science" I feel they are not giving enough credit to the many scientists actively fighting fake news. If I see how someone like Carl T. Bergstrom is fighting fake news on twitter, I think such people deserve some recognition.

I am in doubt of the value of the figure in Box 1. I think this article is really strong. I greatly

enjoyed reading it. This figure is by far the weakest part in the manuscript. It's not needed, overly simplistic, and the corona virus might be a good hook, but this perspective piece will stay relevant in future crises. I would probably take it out myself. If not, the picture might need a credit (I assume the corona virus picture was taken from somewhere).

It was a pleasure to read this perspective. I look forward to seeing it in print, and circulating it among my colleagues.

Signed,
Daniel Lakens

Reviewer: 3

Comments to the Author(s)

This article highlights the importance of "trust in science", the need for a scientifically literate population and for a strong and fruitful relationship between science and policy. It emphasises the dangers of ignoring science based evidence as in climate change denial and the "anti-vaxxer" movement (which might be discussed in the article). Although all these points have been made before, they are well worth highlighting and the article should be published.

I have two concerns which I would like the authors to consider. First, I think there is an unduly pessimistic tone to the article. It refers to re-establishing trust" and whereas I accept that there are issues here, I think there is still a great deal of confidence and respect for science and scientists in many countries and is some (eg the UK), we have seen an increase in the recognition by governments of the importance of science to the economy and society generally. I think the impact of the article will be greater if these positive aspects are recognised more clearly.

Secondly, the style of the article deserves some attention. It could in parts be clearer and more readable.

Author's Response to Decision Letter for (RSOS-200554.R0)

See Appendix A.

Decision letter (RSOS-200554.R1)

Dear Dr Matlin,

It is a pleasure to accept your manuscript entitled "Realigning science, society and policy in uncertain times" in its current form for publication in Royal Society Open Science.

You can expect to receive a proof of your article in the near future. Please contact the editorial office (openscience_proofs@royalsociety.org) and the production office

(openscience@royalsociety.org) to let us know if you are likely to be away from e-mail contact -- if you are going to be away, please nominate a co-author (if available) to manage the proofing process, and ensure they are copied into your email to the journal.

on behalf of Professor Matjaz Perc (Associate Editor)
openscience@royalsociety.org

Appendix A

We are delighted to note the overall views of the three reviewers that the article *'makes very good sense'*, *'is very strong'* and makes points that are *'well worth highlighting'* and that it has been accepted for publication *'subject to minor revisions'*. We are very grateful to the three reviewers and editorial team for their comments and suggestions, the adoption of which we believe has helped to further strengthen the Perspective.

Our responses to specific points made by the reviewers are as follows (highlighted in yellow):

Reviewer 1:

R1-1. "...does not deal in specifics (except for the Covid-19 mention). It's all generalities."

Since the authors' intention was to comment on a very broad set of relationships between science, society and policy without making the Perspective too long, the temptation to give many detailed examples and references to support the arguments was deliberately avoided at the outset – the exception being Covid-19, which was taken as an example of the failures seen in many other cases.

Nevertheless, some further specific examples have been cited in response to this reviewer and to comments and requests by other reviewers for additional information, as highlighted in the responses below.

R1-2. "...fails to pose and refer the reader to crucial questions. For instance..."

In a Perspective of global scope, we had not wished to focus at length on any one country. The reviewer is correct in identifying a number of anomalies and paradoxes in the USA's attitudes to science and responses to the Covid-19 pandemic. There are many complex social, political, economic and cultural factors involved that have played out differently at different points in time. The case of the response to atmospheric ozone depletion is an example of this complexity, involving an interplay of many national and international factors including commercial interests. We have introduced new material and references at the end of Section 3 to try to point the reader to the range and complexity of these issues as succinctly as possible.

Reviewer 2:

R2-1. "Section 4 starts with "The decline in trust and respect and sometimes undermining of the effective functioning of science". However, public trust in science in general has remained stable over the last decades (e.g., <https://www.pewresearch.org/fact-tank/2019/03/22/public-confidence-in-scientists-has-remained-stable-for-decades/>). If the authors want to argue for a decline in trust in science, they will either need to support their argument with data and references, or they might want to be more specific in what they mean (e.g., a greater concern about trust in science, or more visibility of distrust in science – but general trust seems stable. The link goes to 1 survey, but I had a post-doc once collect these surveys across countries – the pattern is basically the same everywhere. Now the questions in these surveys might be too general, and not measure the same type of trust as the authors mean. But **I think some clarification is required.**"

See also **R3-2** for a similar point.

We take the points, as made by Reviewers 2 and 3, that the degrees and locations of the decline in trust in science to which we refer are variable. We have therefore modulated the language and added clarifying text and references to make this explicit, as follows:

- In Section 1: Summary, we have stated that it affects "some sections of society and some policy-makers"; and that "Scientists must strive collectively to re-establish trust by society and politicians where it is damaged, and reinforce conviction of science's central importance in underpinning policy."
- In Section 3, end of [new] first paragraph, addition of the sentence:
This has been widely recognised in, for example, the science and innovation policies of OECD countries. [12]
- In Section 4, 1st paragraph has been rewritten and the two paragraphs introduced in its place summarise some of the current state of knowledge on trust in science:

R2-2. "Personally, I found the last sentence of section 4 too strong: "The culture of science denial displayed by some politicians and sections of society affects the approach to looming planetary disasters and also spawns a social attitude that it is acceptable to scorn rigorously proven evidence and to treat fake science and faked evidence as if it is of at least equal value for decision-making." I mean, yes, these are troublesome developments, but not unique our history. We are relatively far removed from treating fake science as a good source of decision making. The manuscript does not mention there is a huge pushback to fake news as well, so in some ways it also increases the awareness of the importance of reliable information among

others, who work actively to make sure internet pages are reliable (e.g., think of hundreds of thousands of moderators cleaning up internet for a such as reddit).”

We have

- a. referenced the ‘culture of science denial’ displayed by some politicians [Section 4] and
- b. revised the beginning of Section 5 to acknowledge that there has been some pushback (see response to R2-3 & R2-4 below).

R2-3 “In section 5, I would like to see a reference after “Responses by the scientific community to the growing challenges have frequently been muted.” I wonder which examples the authors are thinking of.”

and

R2- 4 “Related to 2 points above, when the authors write “Scientists have often been slow to react publicly to the circulation of fake science” I feel they are not giving enough credit to the many scientists actively fighting fake news. If I see how someone like Carl T. Bergstrom is fighting fake news on twitter, I think such people deserve some recognition.”

This paragraph at the beginning of Section 5 has been rewritten and references introduced to modulate the description of responses, reference the range of responses and provide some of the possible reasons for them.

R2-5 “I am in doubt of the value of the figure in Box 1. I think this article is really strong. I greatly enjoyed reading it. This figure is by far the weakest part in the manuscript. It’s not needed, overly simplistic, and the corona virus might be a good hook, but this perspective piece will stay relevant in future crises. I would probably take it out myself. If not, the picture might need a credit (I assume the corona virus picture was taken from somewhere).”

We have adopted the Reviewer’s suggestion and deleted the Figure in Box 1 and made small consequent changes to the wording to remove reference to a ‘trivalent’ vaccine.

Reviewer 3:

R3-1 “This article highlights the importance of "trust in science", the need for a scientifically literate population and for a strong and fruitful relationship between science and policy. It emphasises the dangers of ignoring science-based evidence as in climate change denial and the "anti-vaxxer" movement (which might be discussed in the article).

We have have adopted the Reviewer’s suggestion and introduced references to climate change denial and the "anti-vaxxer" movement in relation to the discussion of science literacy in the first part of Section 3.

R3-2 “I have two concerns which I would like the authors to consider. First, I think there is an unduly pessimistic tone to the article. It refers to re-establishing trust" and whereas I accept that there are issues here, I think there is still a great deal of confidence and respect for science and scientists in many countries and is some (eg the UK), we have seen an increase in the recognition by governments of the importance of science to the economy and society generally. I think the impact of the article will be greater if these positive aspects are recognised more clearly.

The ‘trust’ addresses the same issue (R2-1) as that made by Reviewer 2 and is dealt with in the response to that point.

We have tried to further emphasise optimism about the role of science and its capacity to solve problems despite the uncertainties, including through the introduction of the quotation (ref 1) and the final paragraph of the article.

R3-3 “Secondly, the style of the article deserves some attention. It could in parts be clearer and more readable.”

We have reviewed tried to clarify some of the denser language and longer sentences and paragraphs – including, in particular, the first paragraph in Section 3 which has now been revised into 3 paragraphs with some re-ordering and references added.